# The Effects of Red Light on Mammalian Sperm Rely upon the Color of the Straw and the Medium Used

**DOI:** 10.3390/ani11010122

**Published:** 2021-01-08

**Authors:** Jaime Catalán, Iván Yánez-Ortiz, Sabrina Gacem, Marion Papas, Sergi Bonet, Joan E. Rodríguez-Gil, Marc Yeste, Jordi Miró

**Affiliations:** 1Equine Reproduction Service, Department of Animal Medicine and Surgery, Faculty of Veterinary Sciences, Autonomous University of Barcelona, E-08193 Bellaterra (Cerdanyola del Vallès), Spain; dr.jcatalan@gmail.com (J.C.); ivan.yanez22@gmail.com (I.Y.-O.); swp.sabrina.gacem@gmail.com (S.G.); papas.marion@gmail.com (M.P.); JuanEnrique.Rodriguez@uab.cat (J.E.R.-G.); 2Biotechnology of Animal and Human Reproduction (TechnoSperm), Institute of Food and Agricultural Technology, University of Girona, E-17003 Girona, Spain; sergi.bonet@udg.edu; 3Unit of Cell Biology, Department of Biology, Faculty of Sciences, University of Girona, E-17003 Girona, Spain

**Keywords:** horse, sperm, red light irradiation, extender, straw

## Abstract

**Simple Summary:**

Several studies have shown that the exposure of semen to red light improves sperm quality and fertilizing ability, which could improve the efficiency of assisted reproductive techniques with irradiated semen. However, despite being considered as possible sources of variation, the effects of the color of the container (straws) or the medium have not yet been evaluated. In this study, 13 ejaculates from different stallions were split into equal fractions, diluted either with Kenney or Equiplus extender, and subsequently packed into straws of five different colors. After storage at 4 °C for 24 h, the sperm were irradiated and different variables, including sperm motility, plasma membrane integrity, and mitochondrial membrane potential, were evaluated. Our results confirm that irradiation increases some motion characteristics and mitochondrial membrane potential without affecting sperm viability and demonstrate that the effects depend on the color of the straw and the extender used.

**Abstract:**

Previous research has determined that irradiation of mammalian sperm with red light increases motility, mitochondrial activity, and fertilization capacity. In spite of this, no study has considered the potential influence of the color of the straw and the extender used. Therefore, this study tests the hypothesis that the response of mammalian sperm to red light is influenced by the color of the straw and the turbidity/composition of the extender. Using the horse as a model, 13 ejaculates from 13 stallions were split into two equal fractions, diluted with Kenney or Equiplus extender, and stored at 4 °C for 24 h. Thereafter, each diluted fraction was split into five equal aliquots and subsequently packed into 0.5-mL straws of red, blue, yellow, white, or transparent color. Straws were either nonirradiated (control) or irradiated with a light–dark–light pattern of 3–3–3 (i.e., light: 3 min, dark: 3 min; light: 3 min) prior to evaluating sperm motility, acrosome and plasma membrane integrity, mitochondrial membrane potential, and intracellular ROS and calcium levels. Our results showed that irradiation increased some motion variables, mitochondrial membrane potential, and intracellular ROS without affecting the integrities of the plasma membrane and acrosome. Remarkably, the extent of those changes varied with the color of the straw and the extender used; the effects of irradiation were more apparent when sperm were diluted with Equiplus extender and packed into red-colored straws or when samples were diluted with Kenney extender and packed into transparent straws. As the increase in sperm motility and intracellular ROS levels was parallel to that of mitochondrial activity, we suggest that the impact of red light on sperm function relies upon the specific rates of energy provided to the mitochondria, which, in turn, vary with the color of the straw and the turbidity/composition of the extender.

## 1. Introduction

Artificial insemination (AI) is a tool widely used today for horse breeding, especially when looking for genetic improvement [1]. The increasing use of this technology, both in the horse and other species, has augmented the interest for semen processing techniques and their optimization, aimed at maximizing their survival and fertilization capacity [2,3]. Unfortunately, semen quality often deviates from expectations and leads to unsatisfactory pregnancy rates [1]. In this context, any protocol or procedure that optimizes its use and helps increase reproductive performance should be considered; for this reason, several approaches have been undertaken in recent years [3,4]. One of these approaches is sperm irradiation; in effect, previous research has demonstrated that red light stimulation, either with low-level lasers or light-emitting diodes (LEDs), increases the motility, ability to elicit in vitro capacitation, fertilizing ability, and lifespan of fresh, liquid-stored, and frozen-thawed sperm in fish [5], birds [6], humans [7,8,9,10,11,12], pigs [13,14,15], sheep [16], dogs [17,18], buffalos [19] donkeys [3], and horses [20,21]. In addition to this, recent studies have shown that the increase in sperm motility in response to LED-based red light is concomitant with that of mitochondrial activity in pigs, donkeys, and horses [3,13,20,22].

The mechanisms through which light exerts its effects are not entirely clear. Three potential mechanisms have been surmised to explain the response of mammalian sperm to red light (reviewed in Yeste et al. [23]). The first of these hypotheses is related to the possible influence of light on transient receptor proteins (TRPs) [24,25,26], which reside in sperm plasmalemma and have been purported to participate in the modulation of thermotaxis [27,28]. The second hypothesis is related to the presence of opsins in mammalian spermatozoa, which absorb light from different spectra [23]; despite being mainly related to the response to thermotaxis [28], they could also be involved in the sperm response to light. Finally, mounting evidence supports the third hypothesis, which confers a crucial role on endogenous cellular photosensitizers, especially those present in the mitochondria [22,23,29]. These photosensitizers absorb light from electromagnetic radiation and then ionize and transfer the absorbed energy into adjacent molecules [30]. This increased energy induces a rise in electrochemical mitochondria potential, which may result in an augmentation of ATP and Ca^2+^ levels [23]. In spite of this, it cannot be ruled out that more than one of the mechanisms proposed by these hypotheses are involved in the sperm response to red light [22,29]. 

Previous studies carried out with low-level laser therapy devices and light-emitting diodes (LEDs) have reported an increase in ATP production via the mitochondrial electron chain [31,32] without damaging the irradiated cells [32,33] or the integrity of their DNA [32,34]. Therefore, it has been suggested that light stimulation can have a safe and positive effect on sperm motility and fertilizing ability both in vivo and in vitro [32]. Nevertheless, the sperm response to irradiation has been reported to depend on different factors, including the type (i.e., fresh, cooled-stored or frozen-thawed) and state of the sample [3], the irradiation of the light beam used [32], the time or pattern of exposure [13], and the species [5]. Given the properties of light emission/absorption, other factors such as the color of the straw and choice of extender could also affect the sperm response to red light. However, to the best of our knowledge, no previous study has examined this possibility despite the wide variety of extenders and colors of commercial straws.

Taking the results obtained in the aforementioned studies (especially those conducted with fresh, cooled-stored, and frozen-thawed horse sperm) [20,21] into account, this study aims at determining whether the color of the straw and the extender used affect the response of cooled-stored sperm to LED-based red light (620–630 nm). Our hypothesis is that the effects of red light on horse sperm depend on the color of the straw and the extender.

## 2. Materials and Methods

### 2.1. Suppliers

All reagents used were of analytical grade and were purchased from Boehringer-Mannheim (Mannheim, Germany), Merck (Darmstadt, Germany), and Sigma-Aldrich (Saint Louis, MO, USA). As far as fluorochromes are concerned, unless otherwise stated, all were purchased from Molecular Probes (Thermo Fisher Scientific; Waltham, MA, USA) and were previously prepared with dimethyl sulfoxide (Sigma-Aldrich). Plastic materials were provided by Nunc (Roskilde, Denmark), and empty straws of different colors (transparent, red, white, blue, and yellow) were purchased from Minitüb GmbH (Tiefenbach, Germany).

### 2.2. Animals and Ejaculates

This study included 13 ejaculates from 13 different adult stallions (age: 5–8 years old) with proven fertility. Animals were housed at the Equine Reproduction Service, Autonomous University of Barcelona (Bellaterra, Cerdanyola del Vallès, Spain), which is an EU-approved semen collection center (Authorization code: ES09RS01E) that operates under strict protocols of animal welfare and health control. All animals were semen donors and were collected under CEE health conditions (free of equine arteritis, infectious anemia, and contagious metritis). As indicated in Catalán et al. [21], this Service runs under the rules of the Regional Government of Catalonia, Spain, and no manipulation of the animals other than semen collection was carried out. The study was approved by the Ethics Committee, Autonomous University of Barcelona (Code: CEEAH 1424).

Ejaculates were collected through a Hannover artificial vagina (Minitüb GmbH, Tiefenbach, Germany), and an in-line nylon mesh filter was used to remove the gel fraction. Upon collection, gel-free semen was split into two fractions of equal volume and immediately diluted 1:5 (*v:v*) either in Kenney [35] or Equiplus extender (Minitüb GmbH; Tiefenbach, Germany), which were selected for their different turbidity. The absorbance of these two extenders was evaluated at 625 nm with a spectrophotometer (Biochrom WPA, Lightwave II; Cambridge, UK) and sterilized; ultrafiltered Milli-Q water was used as blank. Absolute absorbance values of the Equiplus and Kenney extenders were 0.090 and >2.5, respectively. Both extenders were preheated to 37 °C, and sperm concentration was adjusted in all cases to 30 × 10^6^ sperm/mL with a Neubauer chamber (Paul Marienfeld GmbH and Co. KG; Lauda-Königshofen, Germany). Following this, sperm motility (Computer Assisted Semen Analysis, CASA), morphology (eosin–nigrosin staining), and plasma membrane integrity (SYBR14/PI) of each sample were evaluated. All samples were confirmed to fulfill the standard thresholds: ≥60% SYBR14^+^/PI^−^ spermatozoa and ≥70% morphologically normal spermatozoa. Thereafter, semen samples were stored in a refrigerator at 4 °C for 24 h.

### 2.3. Experimental Design

After 24 h of storage, samples extended in either Kenney or Equiplus were split and packed into 0.5-mL straws (Minitüb GmbH) of five different colors (blue, red, yellow, white, and transparent); the sperm concentration was maintained at 30 × 10^6^ sperm/mL at all experimental points. Straws were placed within a programmable photoactivation system (MaxiCow; IUL, SA, Barcelona, Spain). In this device, each straw is in contact with a triple-LED configuration system that emits red light (wavelength window: 620 to 630 nm). The apparatus is equipped with software (IUL, SA) that allows the regulation of intensity and time of exposure. In all cases, the intensity was set at 100%.

Straws of different colors containing sperm diluted by both extenders were irradiated with a light–dark–light interval pattern of 3–3–3 min. Nonirradiated samples (control) were also packed into 0.5-mL straws and left for 9 min in the dark, which was the same time used to irradiate the samples. Upon light stimulation, irradiated and nonirradiated samples were transferred into 1.5-mL tubes. Sperm motility was evaluated with a computer-assisted sperm analysis (CASA) system, and plasma membrane and acrosome integrity, mitochondrial membrane potential, intracellular ROS (peroxides and superoxides), and calcium levels were determined through flow cytometry.

### 2.4. Analysis of Sperm Motility

Sperm motility was evaluated using a computer-assisted sperm analysis (CASA) system (Integrated Sperm Analysis System V1.0; Proiser S.L.; Valencia, Spain). In brief, samples were incubated at 38 °C in a water bath for 5 min, and 5 μL of each sperm sample was placed onto a Makler chamber (Sefi Medical Instruments; Haifa, Israel), previously warmed at 38 °C. Samples were then analyzed under a 10× negative phase-contrast objective (Olympus BX41 microscope; Olympus, Tokyo, Japan). A minimum of 1000 sperm cells was counted per analysis. In each evaluation, percentages of total motility (TMOT, %) and progressively motile spermatozoa (PMOT, %) were recorded together with the following kinetic measures: curvilinear velocity (VCL, μm/s), which is the mean path velocity of the sperm head along its actual trajectory; straight-line velocity (VSL, μm/s), which is the mean path velocity of the sperm head along a straight line from its first to its last position; average path velocity (VAP, μm/s), which is the mean velocity of the sperm head along its average trajectory; percentage of linearity (LIN, %), which is the quotient between VSL and VCL multiplied by 100; percentage of straightness (STR, %), which is the quotient between VSL and VAP multiplied by 100; percentage of oscillation (WOB, %), which is the quotient between VAP and VCL multiplied by 100; mean amplitude of lateral head displacement (ALH, μm), which is the mean value of the extreme side-to-side movement of the sperm head in each beat cycle; frequency of head displacement (BCF, Hz), which is the frequency at which the actual sperm trajectory crosses the average path trajectory. 

CASA settings were those recommended by the manufacturer, i.e., frames: 25 images captured per second; particle area >4 and <75 µm^2^; connectivity = 6; minimum number of images to calculate ALH: 10. The cut-off value for motile spermatozoa was VAP ≥ 10 μm/s; for progressively motile spermatozoa, the cut-off value was STR ≥ 75%.

### 2.5. Flow Cytometry

The integrity of sperm plasma membrane (SYBR14/PI), acrosome integrity (PNA-FITC/PI), mitochondrial membrane potential (JC1), and intracellular levels of peroxides (H_2_DCFDA/PI), superoxides (HE/YO-PRO-1), and calcium (Fluo3/PI) were determined through flow cytometry. Samples were stained and evaluated following the protocol described by Prieto-Martínez et al. [36] and adjusted to horse spermatozoa.

Management of the flow cytometer and analysis of the samples were carried out in accordance with the recommendations of the International Society of Cytometry [37]. The flow cytometer used in this study was a Cell Lab Quanta SC™ (Beckman Coulter, Fullerton, CA, USA), and particles were excited with an argon laser (488 nm) at a power of 22 mW. Prior to staining, sperm concentration was adjusted to 1 × 10^6^ sperm/mL. Every day, the electronic volume (EV) channel was calibrated with 10-µm diameter fluorescent beads (Beckman Coulter), following the manufacturer’s instructions. The flow rate was set at 4.17 µL/min, and the analyzer threshold was established to exclude cell aggregates (particles with a diameter >12 µm) and debris (particles with a diameter < 7 µm). Sperm cells were gated on the basis of EV and side scatter (SS) distributions. Three different optical filters were used (FL1 for the analysis of SYBR14, PNA, H_2_DCFDA, Fluo3, and JC1 monomers, detection width: 505–545 nm; FL2 for the analysis of JC1 aggregates, detection width: 560–590 nm; FL3 for the analysis of PI and HE, detection width: 655–685 nm).

Dot plots were examined using Cell Lab Quanta SC™ MPL Analysis Software (version 1.0; Beckman Coulter) and data from PNA/PI, JC1, H_2_DCFDA/PI, HE/YO-PRO-1; Fluo3/PI were corrected using the percentage of nonstained debris particles found in SYBR14/PI staining, as recommended by Petrunkina et al. [38].

#### 2.5.1. Analysis of Plasma Membrane Integrity

Sperm viability (plasma membrane integrity) was assessed using the LIVE/DEAD^®^ Sperm Viability Kit (SYBR14/PI; Molecular Probes, Thermo Fisher Scientific; Waltham, MA, USA), according to the protocol described by Garner and Johnson [39], and adapted to horse spermatozoa. In brief, samples were first incubated with SYBR14 (final concentration: 100 nM) at 38 °C for 10 min, and then with PI (final concentration: 12 µM) at 38 °C for 5 min. Three sperm populations were distinguished: (i) viable spermatozoa emitting green fluorescence (SYBR14^+^/PI^−^), which appeared on the right side of the lower half of the FL1/FL3 dot plots; (ii) nonviable spermatozoa emitting red fluorescence (SYBR14^−^/PI^+^), which appeared on the left side of the upper half of the FL1/FL3 dot plots; (iii) nonviable spermatozoa emitting both green and red fluorescence (SYBR14^+^/PI^+^), which appeared on the right side of the upper half of the FL1/FL3 dot plots. Nonstained particles (SYBR14^−^/PI^−^), which appeared on the left side of the lower half of the FL1/FL3 dot plots, showed EV/SS distributions similar to spermatozoa and were considered non-DNA debris particles. Percentages of nonstained particles were used to correct the percentages of double-negative sperm populations in the other assessments. Spill-over of FL1 into the FL3 channel was compensated (2.45%).

#### 2.5.2. Analysis of Acrosome Integrity

Plasma membrane integrity was evaluated through PNA/PI costaining, following the procedure described for horse spermatozoa by Rathi et al. [40]. With this purpose, spermatozoa were stained with PNA conjugated with FITC (final concentration: 5 μg/mL) and PI (final concentration: 12 μm) and incubated at 38 °C for 10 min in the dark. Green fluorescence from PNA was collected through FL1, whereas red fluorescence from PI was collected through FL3. As spermatozoa were not previously permeabilized, they were identified and placed in one of the four following populations: (i) spermatozoa with intact plasma membranes (PNA^−^/PI^−^); (ii) spermatozoa with damaged plasma membranes that presented an acrosome membrane that could not be fully intact (PNA^+^/PI^+^); (iii) spermatozoa with damaged plasma membranes and lost outer acrosome membranes (PNA^−^/PI^+^); (iv) spermatozoa with damaged plasma membranes (PNA^+^/PI^−^). Therefore, after PNA/PI staining, two main categories were detected: (i) spermatozoa with an intact plasma membrane (PNA^−^/PI^−^) and (ii) spermatozoa that had damaged their plasma membrane and/or their acrosome membrane (these were represented by the other three categories: PNA^+^/PI^−^, PNA^+^/PI^+^, PNA^−^/PI^+^). Unstained and single-stained samples were used for setting the EV gain, FL1 and FL3 PMT voltages, and for compensation of PNA spill over into the PI channel (2.45%).

#### 2.5.3. Analysis of Mitochondrial Membrane Potential

Mitochondrial membrane potential (MMP) was determined through incubation with JC1 (5,5′,6,6′-tetrachloro-1,1′,3,3′tetraethyl-benzimidazolylcarbocyanine iodide; final concentration: 0.3 μM) at 38 °C for 30 min in the dark. When MMP is low, JC1 forms monomers emitting green fluorescence (JC1_mon_), which are collected through FL1. When mitochondrial membrane potential is high, JC1 forms aggregates emitting orange fluorescence (JC1_agg_), which are detected through FL2. Three sperm populations were distinguished: (i) spermatozoa with green-stained mitochondria (low MMP), (ii) spermatozoa with orange-stained mitochondria (high MMP), and (iii) spermatozoa with heterogeneous mitochondria, stained both green and orange in the same cell (intermediate MMP). Ratios between FL2 (JC1_agg_) and FL1 fluorescence (JC1_mon_) for each of these sperm populations were also evaluated. Spill-over of FL1 into the FL2 channel was compensated (68.5%). Percentages of debris particles found in SYBR14/PI staining (SYBR14^−^/PI^−^) were subtracted from those of spermatozoa with low MMP, and the percentages of all sperm populations were recalculated.

#### 2.5.4. Analysis of Intracellular ROS Levels: H_2_O_2_ and O_2_^−^

Intracellular ROS levels were determined through two oxidation sensitive fluorescent probes, 2′,7′-dichlorodihydrofluorescein diacetate (H_2_DCFDA) and hydroethidine (HE), which detect hydrogen peroxides (H_2_O_2_) and superoxide anions (·O_2_^−^), respectively [41]. Following a modified procedure from Guthrie and Welch [42], a simultaneous differentiation of viable and nonviable sperm was performed using PI (H_2_DCFDA) or YO-PRO-1 (HE).

In the case of peroxides, spermatozoa were incubated with H_2_DCFDA (final concentration: 200 µM) and PI (final concentration: 12 µM) at room temperature for 30 min in the dark. H_2_DCFDA is a stable, cell-permeable, nonfluorescent probe that is converted into 2′,7′-dichlorofluorescein (DCF) in the presence of H_2_O_2_ [42]. Fluorescence of DCF^+^ was measured through FL1 and that of PI was detected through FL3. Four sperm populations were distinguished: (i) viable spermatozoa with low levels of peroxides (DCF^−^/PI^−^), (ii) viable spermatozoa with high levels of peroxides (DCF^+^/PI^−^), (iii) nonviable spermatozoa with low levels of peroxides (DCF^−^/PI^+^), and (iv) nonviable spermatozoa with high levels of peroxides (DCF^+^/PI^+^). Percentages of debris particles found in SYBR14/PI staining (SYBR14^−^/PI^−^) were subtracted from those of viable spermatozoa with low levels of peroxides (DCF^−^/PI^−^) and the percentages of all sperm populations were recalculated. Spill-over of FL1 into the FL3 channel was compensated (2.45%). Data are shown as corrected percentages of viable spermatozoa with high levels of peroxides (DCF^+^/PI^−^) and the geometric mean of DCF^+^-fluorescence intensity in the DCF^+^/PI^−^ sperm population.

Regarding superoxide anions, samples were incubated with HE (final concentration: 4 µM) and YO-PRO-1 (final concentration: 25 nM) at room temperature for 30 min in the dark [42]. Hydroethidine diffuses freely through the plasma membrane and converts into ethidium (E^+^) in the presence of superoxide anions (O_2_^−^) [43]. Fluorescence of ethidium (E^+^) was detected through FL3 and that of YO-PRO-1 was detected through FL1. Four sperm populations were distinguished: (i) viable spermatozoa with low levels of superoxides (E^−^/YO-PRO-1^−^), (ii) viable spermatozoa with high levels of superoxides (E^+^/YO-PRO-1^−^), (iii) nonviable spermatozoa with low levels of superoxides (E^−^/YO-PRO-1^+^), and (iv) nonviable spermatozoa with high levels of superoxides (E^+^/YO-PRO-1^+^). Percentages of debris particles found in the SYBR14/PI test (SYBR14^−^/PI^−^) were subtracted from those of viable spermatozoa with low levels of superoxides (E^−^/YO-PRO-1^−^) and the percentages of all sperm populations were recalculated. Spill-over of FL3 into the FL1 channel was compensated (5.06%). Data are shown as corrected percentages of viable spermatozoa with high levels of superoxides (E^+^/YO-PRO-1^−^) and the geometric mean of E^+^-fluorescence intensity in the E^+^/YO-PRO-1^−^ sperm population.

#### 2.5.5. Intracellular Calcium Levels

Previous studies found that Fluo3 mainly stains mitochondrial calcium in mammalian sperm [44]. For this reason, we combined this fluorochrome with propidium iodide (Fluo3/PI), as described by Kadirvel et al. [45]; the following four populations were identified: (i) viable spermatozoa with low levels of intracellular calcium (Fluo3^−^/PI^−^), (ii) viable spermatozoa with high levels of intracellular calcium (Fluo3^+^/PI^−^), (iii) nonviable spermatozoa with low levels of intracellular calcium (Fluo3^−^/PI^+^), and (iv) nonviable spermatozoa with high levels of intracellular calcium (Fluo3^+^/PI^+^). FL1 spill-over into the FL3 channel (2.45%) and FL3 spill-over into the FL1 channel (28.72%) were compensated.

### 2.6. Statistical Analyses

Statistical analyses were conducted using a statistical package (SPSS^®^ Ver. 25.0 for Windows; IBM Corp., Armonk, NY, USA). Data were first tested for normal distribution (Shapiro–Wilk test) and homogeneity of variances (Levene test), and, if required, they were transformed with arcsin √x. The effects of the color of the straw and the extender on the response of horse sperm to red light were tested with a two-way analysis of variance (ANOVA), followed by a post hoc Sidak test. Sperm motility measures; percentages of spermatozoa with an intact plasma membrane (SYBR14^+^/PI^−^), acrosome-intact spermatozoa (PNA-FITC^−^/PI^−^), spermatozoa with high and intermediate mitochondrial membrane potential, viable spermatozoa with high intracellular calcium levels (Fluo3^+^/PI^−^), viable spermatozoa with high superoxide levels (E^+^/YO-PRO-1^−^), and viable spermatozoa with high peroxide levels (DCF^+^/PI^−^); and geometric mean fluorescence intensities (GMFI) of JC1_agg_, Fluo3^+^, E^+^, and DCF^+^ were analyzed.

Motile sperm subpopulations were determined through the protocol described in Luna et al. [46]. In brief, individual kinematic variables (VCL, VSL, VAP, LIN, STR, WOB, ALH, and BCF) recorded for each spermatozoon were used as independent variables in a principal component analysis (PCA). Kinematic measures were sorted into PCA components, and the obtained matrix was subsequently rotated using the Varimax method with Kaiser normalization. As a result, each sperm cell was assigned a regression score for each of the new PCA components, and these values were subsequently used to run a two-step cluster analysis based on the log-likelihood distance and Schwarz’s Bayesian criterion. Four sperm subpopulations were identified, and each individual spermatozoon was assigned to one of these subpopulations (SP1, SP2, SP3, or SP4). Following this, percentages of spermatozoa belonging to each subpopulation were calculated per sample and used to determine the effects of the color of the straw and the extender on the response of horse sperm to red light through two-way ANOVA and Sidak’s post hoc test. 

In all analyses, the level of significance was set at *p* ≤ 0.05. Data are shown as mean ± standard error of the mean (SEM).

## 3. Results

As expected, no differences in variables were observed between the straws of different colors in the nonirradiated group. For this reason, and in order to simplify the presentation of data, all these results have been grouped and identified as “control” (nonirradiated samples).

### 3.1. Plasma Membrane Integrity

Percentages of membrane-intact spermatozoa (Appendix A) did not differ between nonirradiated and irradiated samples. In addition, neither the color of the straw nor the type of extender had any effect on the percentages of membrane-intact spermatozoa in irradiated and nonirradiated samples (e.g., nonirradiated sperm in Equiplus extender: 46.2% ± 3.9% vs. sperm diluted in Equiplus, packed into blue straws, and irradiated: 47.9% ± 3.6% vs. sperm diluted in Kenney, packed into blue straws, and irradiated: 38.6% ± 3.0%).

### 3.2. Acrosomal Integrity

In a similar fashion to that observed for plasma membrane integrity, percentages of acrosomal-intact spermatozoa (Appendix A) did not differ between nonirradiated and irradiated samples, regardless of the color of the straw or the extender (e.g., nonirradiated sperm in Kenney extender: 39.2% ± 3.1% vs. sperm diluted in Kenney, packed into red straws, and irradiated: 39.7% ± 3.2% vs. sperm diluted in Equiplus, packed into red straws, and irradiated: 49.8% ± 3.6%).

### 3.3. Sperm Motility

No significant differences were observed in the percentages of sperm with total (Appendix A) and progressive motility (Appendix A) between nonirradiated and irradiated samples when compared within each extender. In addition, neither the color of the straw nor the type of extender had any effect on the percentages of sperm with total and progressive motility when the two extenders were compared within each of the colored straws (e.g., total motility: nonirradiated sperm in Equiplus: 56.0% ± 4.5% vs. sperm diluted in Equiplus, packed into yellow straws, and irradiated: 56.5% ± 3.8% vs. sperm diluted in Kenney, packed into yellow straws, and irradiated: 56.3% ± 3.4%; progressive motility: nonirradiated sperm diluted in Kenney: 27.1% ± 2.4% vs. sperm diluted in Kenney, packed into transparent straws, and irradiated: 31.9% ± 2.9% vs. sperm diluted in Equiplus, packed into transparent straws, and irradiated: 28.4% ± 2.6%).

Regarding sperm kinetic variables (Table 1), VCL, VSL, and VAP were significantly (*p* < 0.05) higher in samples diluted in Equiplus and packed into red straws than in their respective control (nonirradiated samples). In addition, VCL and VAP were significantly (*p* < 0.05) higher in sperm diluted in Kenney extender and packed into transparent straws than in the nonirradiated control. In addition to this, STR in samples packed into blue straws and irradiated was significantly higher (*p* < 0.05) in sperm diluted in Kenney than in those diluted in Equiplus extender. 

As shown in Table 2, four different motile sperm subpopulations were identified (SP1, SP2, SP3, and SP4); SP1 was characterized as the fastest subpopulation since it exhibited the highest values in VCL, VSL, and VAP. SP2 was the slowest sperm subpopulation. SP3, although characterized by intermediate speed values (but lower than SP1 and SP4) and LIN and STR values similar to SP1, was the one that showed the highest BCF. Finally, SP4 was characterized by intermediate speed values, which were higher than in SP3, but it was the least linear.

Figure 1a shows the percentages of sperm belonging to SP1. Compared to their respective controls, these percentages were significantly (*p* < 0.05) higher in irradiated samples diluted in Equiplus extender and packed into blue, yellow, or red straws and in irradiated samples diluted in Kenney extender and packed into transparent straws. Percentages of sperm belonging to SP2 were significantly (*p* < 0.05) higher in the control than in irradiated samples diluted in Equiplus extender and packed into yellow, red, or transparent straws (Figure 1b). On the contrary, no significant differences (*p* > 0.05) between nonirradiated and irradiated samples were observed for SP3 and SP4 (Figure 1c,d).

### 3.4. Mitochondrial Membrane Potential

As shown in Figure 2a and Appendix A, percentages of sperm with high MMP were significantly (*p* < 0.05) higher in samples diluted in Equiplus and packed into yellow, red, and transparent straws and in those diluted in Kenney and packed into transparent straws than in their respective controls. In contrast, no significant differences between extenders were observed when nonirradiated and irradiated samples packed into straws of different color were compared. With regard to the percentages of sperm with intermediate MMP, no significant differences between nonirradiated and irradiated samples were observed, regardless of the color of the straw and the extender (Figure 2b).

No significant differences in the geometric mean of JC1_agg_ intensity (orange, FL2) of sperm populations with high (Figure 2c) and intermediate MMP (Figure 2d) were observed between nonirradiated and irradiated samples, regardless of the color of the straw and the extender used. However, the geometric mean of JC1_agg_ intensity (orange, FL2) of the sperm population with a high MMP (Figure 2c) was significantly (*p* < 0.05) higher in samples diluted in Kenney extender and nonirradiated (control) or packed into blue, yellow, red, white, or transparent straws than in their counterparts diluted in Equiplus extender. In addition, the geometric mean of JC1_agg_ intensity (orange, FL2) of the sperm population, with an intermediate MMP in nonirradiated samples (control), was significantly (*p* < 0.05) higher when they were diluted in Kenney than when they were diluted in Equiplus extenders (Figure 2d).

Finally, we also evaluated JC1_agg_/JC1_mon_ ratios of sperm populations with high (Figure 2e) and intermediate MMP (Figure 2f). No significant differences (*p* > 0.05) were observed when comparing nonirradiated and irradiated samples, regardless of the color of the straw and extender used, either within the same diluent or when comparing the two extenders.

### 3.5. Intracellular Peroxide and Superoxide Levels

Figure 3a shows the percentage of viable sperm with high peroxide levels. No significant differences between nonirradiated and irradiated samples were observed, regardless of the color of the straw or the extender used. However, as Figure 3b shows, GMFI of DCF^+^ in the population of viable sperm with high levels of peroxides (DCF^+^/PI^−^) was significantly higher (*p* < 0.05) in transparent, irradiated straws diluted in Equiplus extender than in their respective control (i.e., nonirradiated samples diluted in Equiplus) and transparent, irradiated straws diluted in Kenney extender.

As shown in Figure 3c, percentages of viable spermatozoa with high levels of superoxides (E^+^/YO-PRO-1^−^) and GMFI of E^+^ in the population of viable spermatozoa with high levels of superoxide (Figure 3d) did not differ (*p* > 0.05) between irradiated and nonirradiated samples, regardless of the color of the straw and the extender used.

### 3.6. Intracellular Calcium Levels

Percentages of viable sperm with high intracellular calcium levels (Fluo3^+^/PI^−^; Appendix A) did not differ between nonirradiated and irradiated samples, regardless of the color of the straw and the extender used (e.g., nonirradiated samples diluted in Equiplus: 0.5% ± 0.1% vs. sperm diluted in Equiplus, packed into red straws, and irradiated: 0.8% ± 0.1% vs. samples diluted in Kenney, packed into red straws, and irradiated: 0.8% ± 0.2%). Similar results were observed for the GMFI of Fluo3^+^ in the viable sperm population with high intracellular calcium levels (Appendix A; e.g., nonirradiated sperm diluted in Kenney: 4.4 ± 0.2 vs. sperm diluted in Kenney, packed into white straws, and irradiated: 4.1 ± 0.2 vs. sperm diluted in Equiplus, packed into white straws, and irradiated: 4.1 ± 0.1).

## 4. Discussion

The results of this study agree with previous research, as irradiation with LED-based red light was found to modify some sperm motion variables and increase mitochondrial membrane potential and intracellular ROS of horse sperm without affecting the integrity of the plasma membrane and acrosome. The most remarkable and novel finding, however, was that these effects varied with the color of the straw used to pack sperm before irradiation and with the turbidity of the extender.

Regarding the effects on sperm motility, red light stimulation did not affect TMOT or PMOT, regardless of the color of the straw or the type of diluent used, which agrees with the data reported for dogs [17,18], bulls [47], and horses [20,21]. However, other studies found that irradiation of sperm with red light increases total and progressive motility in humans [7,8,9,10,11], buffaloes [19], sheep [4], pigs [13], and donkeys [3]. In evaluating the presence of motile subpopulations of sperm in horse ejaculates, we identified four separate subpopulations. These results are similar to those previously reported for this species [21,48]. In addition to this, we observed that the percentages of sperm belonging to SP1, which was the fastest subpopulation according to VCL, VSL, and VAP, were significantly higher in irradiated samples that were either diluted with Equiplus extender and packed into blue, yellow, and red straws or diluted with Kenney extender and packed into transparent straws. Furthermore, samples diluted in Equiplus extender and packed into red, yellow, and transparent straws showed significantly lower percentages of sperm belonging to SP2 (the slowest subpopulation) than the control. Therefore, our data confirm the results obtained in previous studies, where irradiation with red light was found to modify the structure of motile sperm subpopulations by decreasing the percentages of the slowest sperm subpopulation [21] and increasing those of the fastest one [18,21,22]. Moreover, we observed that light-stimulation increased some kinetic measures, which agrees with the data reported for other species such as humans [9,34], dogs [17,18], cattle [47], buffaloes [19], pigs [13], donkeys [3,22], and horses [20,21]. These observed differences reinforce the hypothesis that the effects of red light on spermatozoa depend on the specific irradiation pattern [3,13,22,29] and also differ between species [3,5,20,21,22]. At this point, the increase of VCL, VSL, and VAP observed in sperm diluted in Equiplus extender, packed into red straws, and irradiated and the increase of VCL and VAP found in semen diluted in Kenney extender, packed into transparent straws, and irradiated should be emphasized. Moreover, STR also increased in sperm diluted in Kenney extender, packed into blue straws, and irradiated. All these data suggest that the effects of red light on sperm depend on the color of the straw and the medium used. Based on these results, it is reasonable to surmise that the color of the straw and the turbidity of the extender modify the amount of light/energy that reaches the sperm cells. 

At present, there is no clear explanation of how irradiation affects these sperm motion measures as the exact mechanism(s) through which red light stimulates sperm still remains unknown. However, one of the established hypotheses postulates that red light may boost mitochondrial activity, which could be relevant to explain the effects observed in sperm kinetics. Related to this, our data on the analysis of mitochondrial membrane potential (JC1) would agree with this possibility because there was an increase in the percentages of sperm with high mitochondrial membrane potential in samples packed into transparent and red straws and irradiated, regardless of the extender used (Equiplus or Kenney). This matches with Siqueira et al. [47], who found that irradiation of bovine sperm with a He-Ne laser at a wavelength of 633 nm increases the percentage of sperm cells with high mitochondrial membrane potential, and with Yeste et al. [13], who observed that irradiation with red LED light at a wavelength between 620–630 nm augments the percentages of pig sperm with high mitochondrial membrane potential. All these data suggest that red light stimulation could increase mitochondrial activity through photosensitizers present in the electronic chain, such as cytochrome C [13,22,29,49], which would underlie the increase observed in sperm motility. 

In addition to the aforementioned, because ROS are mainly generated in the mitochondria as a byproduct of the electronic chain and following the previously established hypothesis, which points out that one of the first effects of light on sperm is the production of ROS [5,50], the generation of intracellular peroxide and superoxide levels was also evaluated. While sperm irradiation did not affect superoxide generation, we found an increase in the levels of peroxides in those irradiated after dilution in Equiplus and packing into transparent straws. This rise in intracellular ROS levels agrees with Zan-Bar et al. [5], Catalán et al. [20], and Cohen et al. [50], who suggested that ROS formation would be mediated through specific endogenous cellular photosensitizers such as mitochondrial cytochromes. In this sense, it has been reported that although an excess of ROS production produced by irradiation with light could be detrimental to sperm cells [5,50], low ROS levels are beneficial for sperm motility and fertilizing ability [5]. Whilst more studies are needed to set a relationship between fertilization ability and high mitochondrial membrane potential, intracellular ROS, and sperm motility, H_2_O_2_ has been suggested to be the active molecule involved in the light-mediated changes of sperm fertilizing capacity [50], which is consistent with Zan-Bar et al. [5] and de Lamirande et al. [51], who indicated that low concentrations of ROS participate in the signaling transduction pathways related to sperm capacitation and acrosomal reaction. Therefore, ROS can have both harmful and beneficial effects on sperm, and the delicate balance between the amounts of ROS produced and ROS scavengers at any time point determines whether a particular sperm function parameter is compromised or boosted [50]. In this sense, the extent of increase in intracellular ROS levels observed in this study was not enough to negatively affect sperm motility and viability, which is similar to that reported by Catalán et al. [3] in a study conducted with fresh and cooled-stored donkey semen. This increase in intracellular ROS (peroxides), observed herein after irradiation, together with the variation seen due to the color of the straw and the extender used, was concomitant with a rise in mitochondrial activity. These findings reinforce the conjecture that ROS formation caused by light would be mediated by specific endogenous cellular photosensitizers such as mitochondrial cytochromes. Furthermore, cytochrome complexes are also known to be implicated in the intrinsic apoptotic pathway [52], and both ROS generation and modulation of apoptotic-like changes are crucial to cause and control sperm capacitation [53]. Therefore, red light-induced changes in cytochrome C complex activity could ultimately affect sperm capacitation and survival. Surprisingly, however, our results did not show an increase in intracellular calcium levels, which is a crucial secondary messenger involved in the modulation of sperm motility and capacitation [54,55]. Related to this, it is worth noting that our data differ from those reported in previous studies, where light-stimulation was found to increase intracellular levels of calcium [30,50]. This could be explained by different conditions of time and intensity of radiation between the current study and the others, as previous research indicates that sperm irradiation can have stimulatory or inhibitory effects on calcium transport, depending on the intensity of the light used [56].

Regarding the effects of irradiation on the integrity of the plasma membrane, no significant differences were found between irradiated and nonirradiated samples. These results were similar to those reported by Yeste et al. [13] and Pezo et al. [14] in pigs and by Catalán et al. in horses and donkeys [3,20,22]. Similarly, no negative impact of irradiation on acrosomal integrity was observed, which concurs with previous studies in rabbits [16], pigs [13,57], and donkeys [22]. This supports the idea that under the conditions tested herein, stimulation of sperm with red light is safe and can have a positive effect on sperm motility and mitochondrial membrane potential, in agreement with Gabel et al. [32].

Finally, the differences observed in this study between straw colors and extenders with regard to mitochondrial activity, intracellular levels of peroxides, and motility suggest that these two factors also influence the sperm response to light. In fact, the impact of red light on mammalian sperm has been previously reported to rely on the precise rhythm and intensity of light [13] and the functional status of the cell [3]. In agreement with this and with the hypothesis that light acts on endogenous cellular photosensitizers of mitochondria, it is reasonable to suggest that the energy supplied to the mitochondrial electron chain by red light is proportional to the exposure time and the intensity of the light used. The final consequence of this phenomenon would be that the color of the straw and the opacity/turbidity of the medium influence the intensity of the light that feeds mitochondria, which would generate a different effect on sperm cells.

## 5. Conclusions

Our results confirm that LED-based red light irradiation increases some sperm motion variables, mitochondrial membrane potential, and intracellular ROS without affecting the integrity of the sperm membrane and acrosome. However, these effects vary with the color of the straw and the extender/medium used. Given that increased motility and intracellular ROS levels are concomitant with a rise in mitochondrial activity, we suggest that the impact of irradiation on sperm depends on the precise rates of energy provided by the light that feeds the mitochondria. Remarkably, such an energy rate, sensed by mitochondrial photosensitizers, varies with the color of the straw and the extender/medium used, so that these two aspects have to be taken into consideration when sperm are irradiated. In effect, as could be observed in this study, the greatest effects were obtained in samples diluted in Equiplus extender, packed into red straws, and irradiated and samples diluted in Kenney extender, packed into transparent straws, and irradiated.

## Figures and Tables

**Figure 1 animals-11-00122-f001:**
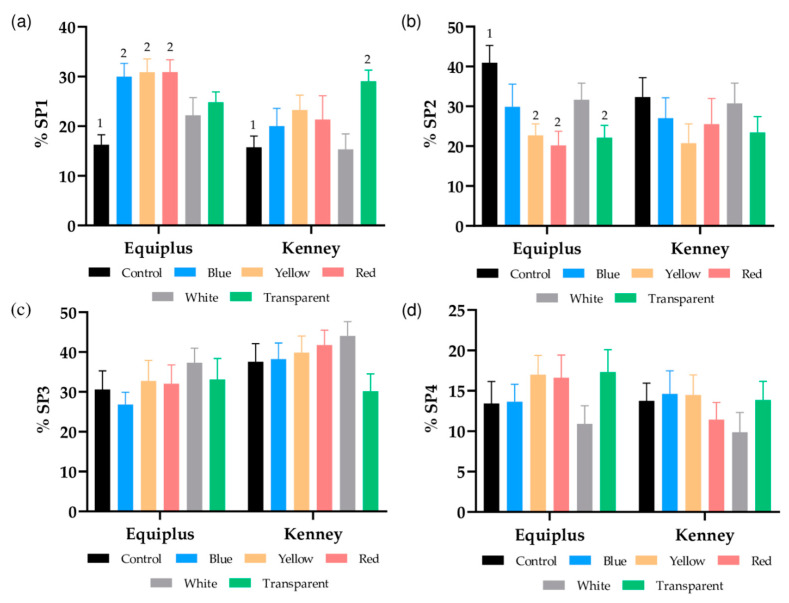
Effects of the color of the straw, extender, and light stimulation on the structure of motile sperm subpopulations in control (nonirradiated) and irradiated samples packed into straws of different color and extended either with Equiplus or Kenney extender. (**a**) Subpopulation 1 (SP1, which was the fastest subpopulation for VCL, VSL and VAP); (**b**) Subpopulation 2 (SP2, the slowest); (**c**) Subpopulation 3 (SP3); (**d**) Subpopulation 4 (SP4). The different numbers (1, 2) indicate significant differences (*p* < 0.05) between irradiated and nonirradiated samples packed into different colored straws within the same diluent. The absence of numbers indicates the lack of statistical differences between irradiated and nonirradiated samples packed into different colored straws within the same diluent. On the other hand, no significant differences between nonirradiated and irradiated samples were observed when the two extenders were compared within the same treatment. Data are shown as mean ±SEM of 13 separate experiments.

**Figure 2 animals-11-00122-f002:**
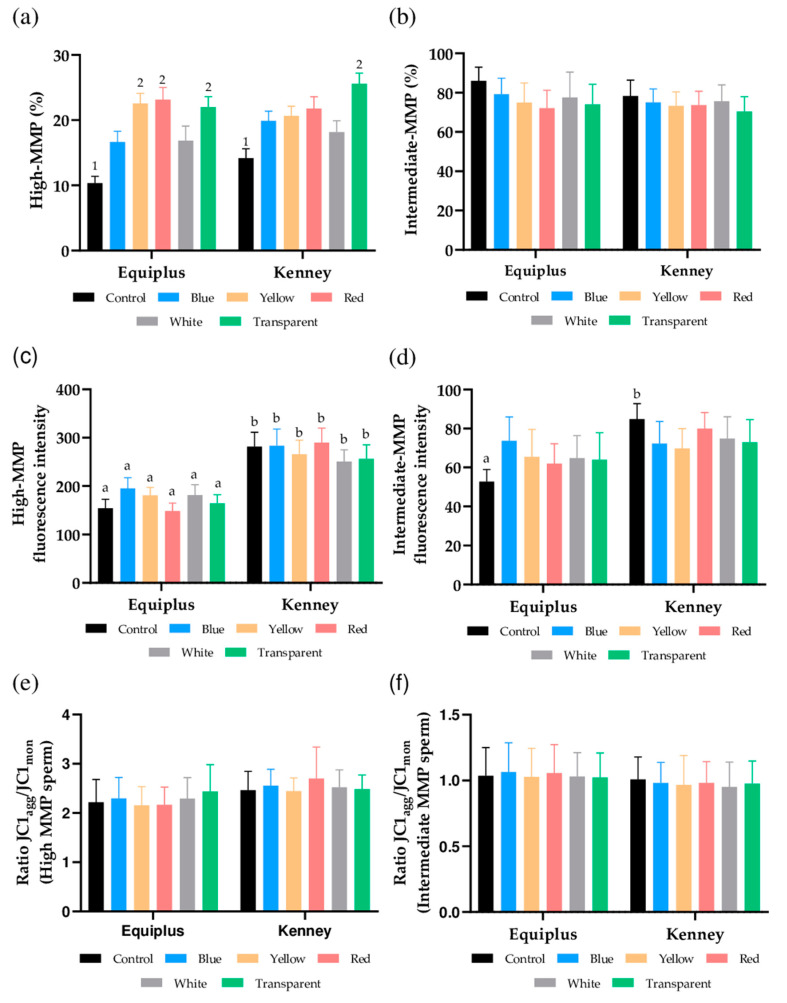
Effects of the color of the straw, extender, and light stimulation on mitochondrial membrane potential in control (nonirradiated) and irradiated samples packed into straws of different color and extended either with Equiplus or Kenney extender. The results are presented as percentages of sperm with high mitochondrial membrane potential (MMP; **a**) and with intermediate mitochondrial membrane potential (MMP; **b**), geometric mean of fluorescence intensity of JC1_agg_ (GMFI, FL2) in sperm populations with high (**c**) and intermediate MMP (**d**), and JC1_agg_/JC1_mon_ ratios (GMFI FL2/GMFI FL1) in sperm populations with high (**e**) and intermediate MMP (**f**) in nonirradiated (control) and irradiated samples. Different numbers (1, 2) indicate significant differences (*p* < 0.05) between nonirradiated and irradiated samples packed into straws of different color within the same diluent. Different letters (a, b) indicate significant differences (*p* < 0.05) between the two extenders within nonirradiated or samples irradiated and packed into straws of different color. The absence of numbers indicates the lack of statistical differences between irradiated and nonirradiated samples within the same diluent, and the absence of letters indicates the lack of differences when comparing a given treatment between both diluents. Data are shown as mean ± SEM of 13 separate experiments.

**Figure 3 animals-11-00122-f003:**
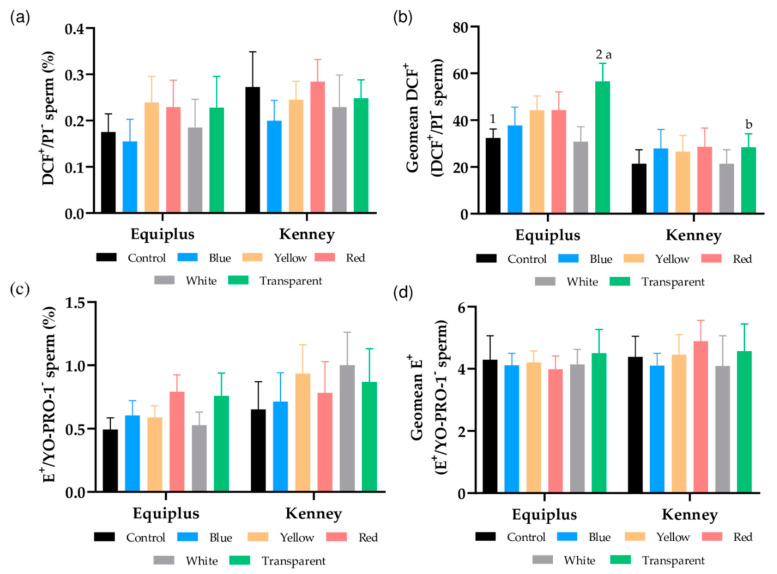
Effects of the color of the straw, extender, and light stimulation on intracellular ROS levels in control (nonirradiated) and irradiated samples packed into straws of different color and extended either with Equiplus or Kenney extender. Data are shown as (**a**) percentages of viable spermatozoa with high peroxide levels (DCF^+^/PI^−^); (**b**) geometric mean of DCF^+^-intensity (GMFI, FL1 channel) in the population of viable spermatozoa with high peroxide levels (DCF^+^/PI^−^); (**c**) percentages of viable spermatozoa with high superoxide levels (E^+^/YO-PRO-1^−^); (**d**) geometric mean of E^+^-intensity (GMFI, FL3 channel) in the population of viable spermatozoa with high superoxide levels (E^+^/YO-PRO-1^−^). Different numbers (1, 2) indicate significant differences (*p* < 0.05) between nonirradiated and irradiated samples packed into straws of different color within the same diluent. Different letters (a, b) indicate significant differences (*p* < 0.05) between the two extenders within nonirradiated or samples irradiated packed into straws of different color. The absence of numbers or letters indicates the lack of statistical difference between irradiated and nonirradiated samples within the same diluent or when comparing a given treatment between Kenny and Equiplus extenders. Data are shown as mean ± SEM of 13 separate experiments.

**Table 1 animals-11-00122-t001:** Effects of the color of the straw (blue, yellow, red, white, and transparent), extender, and light stimulation on sperm kinetic variables in nonirradiated (control) and irradiated samples.

Extender	Treatment (Straw Color)	Kinetic Variables (Mean ± SEM)							
		VCL (µm/s)	VSL (µm/s)	VAP (µm/s)	LIN (%)	STR (%)	WOB (%)	ALH (µm)	BCF (Hz)
**Equiplus**	Control	91.2 ± 3.1 ¹	54.0 ± 2.9	74.2 ± 3.2 ¹	59.3 ± 1.4	73.5 ± 1.9	80.8 ± 1.1	3.0 ± 0.2	8.9 ± 0.2
	Blue	98.5 ± 4.4	60.4 ± 3.6	82.6 ± 4.3	61.2 ± 214	72.9 ± 1.3 ᵃ	83.0 ± 1.4	3.0 ± 0.2	8.9 ± 0.4
	Yellow	102.0 ± 4.7	62.1 ± 4.1	82.8 ± 3.4	61.0 ± 1.7	75.6 ± 1.8	80.7 ± 1.4	3.3 ± 0.2	9.2 ± 0.1
	Red	106.5 ± 3.2 ^2^	68.8 ± 2.6 ²	89.3 ± 3.0 ²	64.7 ± 2.1	77.2 ± 2.9	83.8 ± 0.9	3.3 ± 0.2	8.8 ± 0.2
	White	96.7 ± 4.0	61.3 ± 3.9	81.2 ± 4.3	63.1 ± 1.4	75.5 ± 1.6	83.5 ± 0.8	2.9 ± 0.2	8.7 ± 0.2
	Transparent	102.9 ± 4.9	62.9 ± 3.1	83.6 ± 4.2	62.8 ± 1.8	77.8 ± 2.1	82.7 ± 0.6	3.2 ± 0.2	8.9 ± 0.2
**Kenney**	Control	93.8 ± 2.8 ¹	56.7 ± 3.6	72.3 ± 3.1 ¹	59.9 ± 2.6	78.7 ± 2.2	76.1 ± 2.7	2.9 ± 0.1	10.1 ± 0.4
	Blue	97.5 ± 4.3	63.3 ± 2.8	77.7 ± 4.5	63.8 ± 2.2	82.4 ± 1.6 ᵇ	81.0 ± 2.2	2.8 ± 0.1	9.2 ± 0.4
	Yellow	100.9 ± 4.1	63.4 ± 2.5	80.4 ± 3.9	63.3 ± 2.0	81.2 ± 2.5	78.6 ± 2.3	3.0 ± 0.1	9.9 ± 0.4
	Red	96.1 ± 3.6	62.6 ± 2.5	77.7 ± 4.3	63.7 ± 1.8	81.8 ± 1.8	78.6 ± 2.3	3.0 ± 0.1	9.8 ± 0.5
	White	92.1 ± 3.2	56.0 ± 2.5	70.8 ± 3.7	60.7 ± 2.1	80.1 ± 2.0	76.2 ± 2.5	3.0 ± 0.1	10.4 ± 0.4
	Transparent	107.4 ± 3.0 ^2^	66.3 ± 2.8	86.0 ± 3.0 ^2^	62.0 ± 2.2	78.8 ± 2.4	79.1 ± 2.5	3.1 ± 0.1	9.8 ± 0.5

SEM: standard error of the mean; VCL (μm/s): curvilinear velocity; VSL (μm/s): straight line velocity; VAP (μm/s): average path velocity; LIN (%): linearity; STR (%): straightness; WOB (%): wobble; ALH (μm): amplitude of lateral head displacement; BCF (Hz): beat-cross frequency. Different numbers (^1, 2^) indicate significant differences (*p* < 0.05) between nonirradiated and irradiated samples packed into straws of different color within the same extender (i.e., Kenney and Equiplus). Different letters (^a, b^) indicate significant differences (*p* < 0.05) between extenders within nonirradiated or samples packed into straws of different color. The absence of numbers indicates the lack of statistical differences (*p* > 0.05) between irradiated and nonirradiated samples within the same diluent. The absence of letters means the lack of significant differences between irradiated/nonirradiated sperm between the two extenders. Data are shown as mean ± SEM of 13 separate experiments.

**Table 2 animals-11-00122-t002:** Descriptive parameters (mean ± SEM; range) of the four sperm subpopulations (SP1, SP2, SP3, and SP4) identified in nonirradiated and irradiated samples diluted in the two extenders and packed into straws of different color.

	SP1		SP2		SP3		SP4	
N	10,647		12,622		14,674		8748	
Parameter	Mean ± SEM	Range	Mean ± SEM	Range	Mean ± SEM	Range	Mean ± SEM	Range
VCL (µm/s)	146.7 ± 0.4	108.2–247.3	61.9 ± 0.2	10.0–126.6	94.5 ± 0.1	54.9–149.2	126.3 ± 0.3	71.9–248.3
VSL (µm/s)	108.5 ± 0.2	63.4–198.3	33.4 ± 0.1	4.0–54.3	71.4 ± 0.1	42.6–123.6	40.0 ± 0.2	4.2–112.6
VAP (µm/s)	126.6 ± 0.2	10.2–121.0	45.4 ± 0.1	7.6–80.4	82.5 ± 0.1	48.9–120.0	97.4 ± 0.3	25.1–201.5
LIN (%)	75.0 ± 0.1	10.0–99.5	56.4 ± 0.2	8.3–96.6	76.6 ± 0.1	42.0–99.3	32.0 ± 0.2	2.7–59.8
STR (%)	85.9 ± 0.1	5.3–96.9	74.6 ± 0.2	4.0–95.2	87.1 ± 0.08	52.1–99.7	42.4 ± 0.2	1.5–97.0
WOB (%)	87.2 ± 0.1	37.8–100.0	75.1 ± 0.2	23.6–98.6	88.0 ± 0.09	45.9–100.0	78.0 ± 0.2	19.3–99.2
ALH (µm)	3.9 ± 0.02	0.5–11.6	2.2 ± 0.01	0.2–6.8	2.6 ± 0.01	0.3–7.4	3.9 ± 0.02	1.0–10.3
BCF (Hz)	8.2 ± 0.03	0.0–21.0	8.7 ± 0.03	0.0–22.0	9.1 ± 0.03	0.0–22.0	7.7 ± 0.05	0.0–21.6

SEM: standard error of the mean; VCL (μm/s): curvilinear velocity; VSL (μm/s): straight line velocity; VAP (μm/s): average path velocity; LIN (%): linearity; STR (%): straightness; WOB (%): wobble; ALH (μm): amplitude of lateral head displacement; BCF (Hz): beat-cross frequency.

## Data Availability

The data presented in this study are available on request from the corresponding author.

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
