# Peer review of "The Effects of Red Light on Mammalian Sperm Rely upon the Color of the Straw and the Medium Used"

_animals, 2021, doi:10.3390/ani11010122_

Round 1
Reviewer 1 Report
The manuscript “The effects of red light on mammalian sperm rely upon the color of the recipient and the medium used” evaluates the effects of red light irradiation on horse sperm, focusing on the variation of cell response according to the straw color and extender. The results are interesting and new, showing that indeed the effect of irradiation particularly on motility, mitochondrial membrane potential and ROS production, is influenced by both straw color and extender. The manuscript can be therefore suitable for publication but after revision. Several parameters were evaluated and many groups were compared. As a consequence, the results section is heavy and confusing. Therefore, it is suggested to make an effort to improve the presentation of the results in order to make it clearer to the reader. This may be achieved also by omitting figures not providing useful information (no differences for instance) and limiting superscripts to the actual differences.
Following are specific comments:
L 21-23 It is suggested to change sentence as “However, despite being regarded as potential sources of variation, the effects of the color of the recipient (straws) or the medium have not yet been evaluated”.
L 27-29 Changes as “Our results confirmed that irradiation increases some motility parameters and mitochondrial membrane potential, without affecting sperm viability, and demonstrated that the effects depend on the color of the recipient and the extender used”.
L 41-45. Rephrase the sentence, too long.
L 47 precise?
L 86-87 What do you mean with nature and state of the samples? Isn’it the same meaning? If so just use state of the sample
L 301 Change as “ As expected, in non-irradiated group no differences in all sperm parameters were observed among straws of different colors”.
Figure 2. presentation of results is confusing. I would suggest to leave the superscripts a and b only where the difference is, i.e. white straws Equiplus vs Kenney
L 340 “higher” is missing
Table 1. The table is heavy, it is suggested at least to omit all avoidable superscripts. Just leave those corresponding to differences
Figure 3 can be omitted since there are no differences, you may report the ranges in the text.
Figure 4 and 5. As mentioned before, attention needs to be paid with the use of superscripts that make results not easy to read
L 440 Change “increased” with “increase”
L 462-464 Rephrase the sentence please
L 464-466 Suggested to change as “In this study increased VSL, VAP and LIN were observed in samples irradiated, packed into red straws and diluted with the Equiplus extender and in transparent straws diluted with Kenney extender”
L468 “the increase….suggests”. The effect of color straws is evident but changes of different examined parameters depending on color may be better discussed. For instance “why VSL, VAP and LIN increase in red straws (Equiplus) and transparent (Kenney), while STR increases in blue straw (Kenney) and progressive motility in white (equiplus)?
Discussion
The irradiation resulted in increased motility, mitochondrial membrane potential and ROS, depending on color straws and extender. This has been stated, with a brief mention to the potential deleterious effects of ROS on cells. As membrane and acrosome integrity are unaffected, the authors conclude that it is likely that ROS increased to an extend that is not harmful but rather beneficial. Are there other studies to cite where high mitochondrial membrane potential and ROS together with motility have been associated to fertilizing ability??? If yes, please mention. Otherwise, I would suggest to mention that fertilizing ability should be evaluated before drawing conclusions. Finally, the conclusions are that there is an effect of color straws and extender on cell response to red light. I agree with this but according to the results what the authors suggest is not stated. Which are the good combinations should be reported in the conclusions. Equiplus extender and red straws? Leave a take home message to the reader.
Author Response
Reviewer #1
General comment: The manuscript “The effects of red light on mammalian sperm rely upon the color of the recipient and the medium used” evaluates the effects of red light irradiation on horse sperm, focusing on the variation of cell response according to the straw color and extender. The results are interesting and new, showing that indeed the effect of irradiation particularly on motility, mitochondrial membrane potential and ROS production, is influenced by both straw color and extender. The manuscript can be therefore suitable for publication but after revision. Several parameters were evaluated and many groups were compared. As a consequence, the results section is heavy and confusing. Therefore, it is suggested to make an effort to improve the presentation of the results in order to make it clearer to the reader. This may be achieved also by omitting figures not providing useful information (no differences for instance) and limiting superscripts to the actual differences.
Answer: We really appreciate this positive feedback and would like to thank the reviewer for their kind review of our manuscript and for giving us such an extensive feedback. We made the suggested changes in relation to Figures and superscripts, as well as answered the questions posed by the reviewer. Thank you for these comments.
Following are specific comments:
Comment 1: L 21-23 It is suggested to change sentence as “However, despite being regarded as potential sources of variation, the effects of the color of the recipient (straws) or the medium have not yet been evaluated”
Answer: Thank you very much for your comment. We have revised this sentence and made the change suggested by the reviewer.
Comment 3: L 27-29 Changes as “Our results confirmed that irradiation increases some motility parameters and mitochondrial membrane potential, without affecting sperm viability, and demonstrated that the effects depend on the color of the recipient and the extender used”.
Answer: Thank you very much for your recommendation. We have revised this sentence and made the change as suggested by the reviewer.
Comment 4: L 41-45. Rephrase the sentence, too long.
Answer: Thank you very much for this recommendation. We have revised this sentence and made the change suggested by the reviewer.
Comment 5: L 47 precise?
Answer: Thank you very much for your question. This means that based on the results obtained in this study, we suggest that the impact of red light on sperm function depends on the specific (precise) rates of energy that light provides to the mitochondria, which vary with time, container color and the turbidity / composition of the diluent. We have revised this sentence and modified it to facilitate understanding.
Comment 6: L 86-87 What do you mean with nature and state of the samples? Isn’it the same meaning? If so just use state of the sample
Answer: Thank you very much for your question. Nature refers to the type of sample, that is, whether it is a fresh, chilled or frozen-thawed semen sample. Instead, status refers to the condition of the semen sample or quality. This has been corrected in the text to avoid confusion.
Comment 7: L 301 Change as “As expected, in non-irradiated group no differences in all sperm parameters were observed among straws of different colors”.
Answer: Thank you very much for your recommendation. We have revised this sentence and made the change suggested by the reviewer.
Comment 8: Figure 2. presentation of results is confusing. I would suggest to leave the superscripts a and b only where the difference is, i.e. white straws Equiplus vs Kenney
Answer: Thank you very much for this comment and suggestion, however, since the increase in the number of replicates led the results to change, this Figure has been removed from the Manuscript body and moved to supplementary material.
Comment 9: L 340 “higher” is missing
Answer: Thank you very much for your recommendation. We have revised this sentence and made the change suggested by the reviewer.
Comment 10: Table 1. The table is heavy, it is suggested at least to omit all avoidable superscripts. Just leave those corresponding to differences.
Answer: Thank you very much for your recommendation, which was considered and the table modified accordingly.
Comment 11: Figure 3 can be omitted since there are no differences, you may report the ranges in the text.
Answer: Thank you very much for this suggestion. However, by increasing the number of replicates from 7 to 13 animals, significant differences were observed in this figure. Therefore, both Results and Discussion have been modified accordingly.
Comment 12: Figure 4 and 5. As mentioned before, attention needs to be paid with the use of superscripts that make results not easy to read.
Answer: Thank you very much for your comment, the suggestion made by the reviewer was considered and superscripts were only used when there were significant differences between treatments.
Comment 13: L 440 Change “increased” with “increase”
Answer: Thank you very much for this recommendation. We have revised this sentence and made the change suggested by the reviewer.
Comment 14: L 462-464 Rephrase the sentence please
Answer: Thank you very much for this recommendation. We have revised this sentence and made the change suggested by the reviewer.
Comment 15: L 464-466 Suggested to change as “In this study increased VSL, VAP and LIN were observed in samples irradiated, packed into red straws and diluted with the Equiplus extender and in transparent straws diluted with Kenney extender”
Answer: Thank you very much for this recommendation. We have revised this sentence and made the change suggested by the reviewer.
Comment 16: L468 “the increase….suggests”. The effect of color straws is evident but changes of different examined parameters depending on color may be better discussed. For instance “why VSL, VAP and LIN increase in red straws (Equiplus) and transparent (Kenney), while STR increases in blue straw (Kenney) and progressive motility in white (equiplus)?
Answer: Thank you very much for this suggestion, it was considered and added to the discussion.
Comment 17: Discussion. The irradiation resulted in increased motility, mitochondrial membrane potential and ROS, depending on color straws and extender. This has been stated, with a brief mention to the potential deleterious effects of ROS on cells. As membrane and acrosome integrity are unaffected, the authors conclude that it is likely that ROS increased to an extend that is not harmful but rather beneficial. Are there other studies to cite where high mitochondrial membrane potential and ROS together with motility have been associated to fertilizing ability??? If yes, please mention. Otherwise, I would suggest to mention that fertilizing ability should be evaluated before drawing conclusions. Finally, the conclusions are that there is an effect of color straws and extender on cell response to red light. I agree with this but according to the results what the authors suggest is not stated. Which are the good combinations should be reported in the conclusions. Equiplus extender and red straws? Leave a take home message to the reader.
Answer: Thank you very much for your suggestions, these were considered and added to the discussion and conclusions.
Reviewer 2 Report
In the manuscript, the author analyzed plasma membrane integrity, acrosomal integrity, sperm motility, MMP, ROS, calcium levels. As the author said that transient receptor proteins (TRP) reside in sperm plasmalemma. It is recommended that the authors examine TRP expression to clarify the mechanism of red light. There are no significant differences in most indicators. It is recommended that the author analyses more sperm indicators to clarify the effect of red light on sperm. In semen preservation, elevated sperm motility, ROS levels and mitochondrial activity may not be conducive to semen preservation. In addition to the histogram data, the author should provide the necessary pictures for some results. The number of repeated experiments should be increased to improve the reliability of the results.
Author Response
Reviewer #2
General comment: In the manuscript, the author analyzed plasma membrane integrity, acrosomal integrity, sperm motility, MMP, ROS, calcium levels. As the author said that transient receptor proteins (TRP) reside in sperm plasmalemma. It is recommended that the authors examine TRP expression to clarify the mechanism of red light. There are no significant differences in most indicators. It is recommended that the author analyses more sperm indicators to clarify the effect of red light on sperm. In semen preservation, elevated sperm motility, ROS levels and mitochondrial activity may not be conducive to semen preservation. In addition to the histogram data, the author should provide the necessary pictures for some results. The number of repeated experiments should be increased to improve the reliability of the results.
Answer: We appreciate the reviewer’s feedback and consideration, and we have taken into consideration their comments. First of all, it is important to mention that our work team has carried out a series of studies on the effects of irradiation on mammalian sperm. These studies aimed to determine the most appropriate irradiation protocols in different species, the positive and / or negative effects of red light irradiation on sperm, both in vivo and in vitro, and the possible mechanisms of action of red light on sperm. Therefore, while we share with the reviewer that it would be interesting to evaluate the expression of TRP to address its relationship with red-light in horse sperm, this is out of the scope of this Manuscript. In addition, while performing a greater number of tests would be interesting, we think that the current data are enough to reach the conclusions of this work. Furthermore, this study complements other research conducted by our team in both the horse and other species. We would also like to emphasize that the objective of this work was to determine whether the effects of red-light stimulation on equine semen rely on the color of the straw or the type of diluent used, since although previous studies carried out by our group already determined that light-irradiation did affect both fresh and frozen-thawed sperm in this species, they did not investigate whether the color of the straw or the diluent used had any impact. On the other hand, and following the reviewer’s request, we have added representative flow cytometry images for some results. In addition to this, the number of repeated experiments was increased to improve the reliability of the results.
Reviewer 3 Report
General comments: This manuscript presents data from multiple ejaculates from 7 equine stallions to determine effects of two types of semen extenders and five colors of semen straws in response to irradiation (or not) on measures of sperm motility, acrosome and plasma membrane integrity, mitochondrial membrane potential, intracellular reactive oxygen species, and intracellular calcium levels. The manuscript is very thorough and generally well written but could be improved with less emphasis on results that were not significantly different and more clarification and emphasis for findings that were significant.
Specific comments:
L 20 and throughout the manuscript: The term “parameters” is used frequently and in most cases more appropriate terms would be “variables” or “measures.” “Parameters” typically would be used to describe characteristics of variables such as means, standard errors, etc.
L 26, 37, 62, 64, 104, 112,131, 142, 287, 294, and elsewhere as appropriate: Use commas after each term in a series of terms including after the penultimate term.
L 29 and elsewhere: The term “recipient” is not common from my perspective and “straw” should be substituted consistently throughout the manuscript.
L 45, 475, 485: Use “B(b)ecause” rather than “S(s)ince.”
L 56: Delete “In addition to this” and start sentence with “Unfortunately, …”
L 57-60: Too wordy, rewrite.
L 107: Include range of age of stallions if known.
L 274: Use plural “Statistical analyses”
L 286-297, L 349-353, Table 2 and Figure 3: A lot of effort went into defining and assigning individual spermatozoa into 4 subpopulations (SP1, SP2, SP3, and SP4) with SP3 and SP4 being defined but SP1 and SP2 were not specifically described. Regardless, there were no significant findings and Figure 3 is not needed; ranges of mean percentages with standard errors can be reported in the text. Table 2 may not be needed either but the data included may be of interest?
L 301: Use “among” rather than “between.”
Figure 1 includes no significant findings and ranges of percentages with standard errors could be simply reported in the text.
Figure 2 includes only one significant result and that could be important or could be an outlier. Therefore, that Figure may not be needed either and the significant difference could be handled in the text and discussion.
Table 1 does not stand alone and not easily interpreted without referring to descriptions in the text. Each of the terms needs to be defined specifically in the table footnotes. Similar suggestion for Table 2 if it is kept.
Figures 4 and 5 need to include clear definition of terms used in the figure legends so they can be interpreted without referring to the text.
Figure 6 has no significant findings and those data can be reported as ranges with standard errors in the text.
Author Response
Reviewer #3
General comments: This manuscript presents data from multiple ejaculates from 7 equine stallions to determine effects of two types of semen extenders and five colors of semen straws in response to irradiation (or not) on measures of sperm motility, acrosome and plasma membrane integrity, mitochondrial membrane potential, intracellular reactive oxygen species, and intracellular calcium levels. The manuscript is very thorough and generally well written but could be improved with less emphasis on results that were not significantly different and more clarification and emphasis for findings that were significant.
Answer: We really appreciate this positive feedback and would like to thank the reviewer for their kind review of our Manuscript and for giving us such an extensive feedback. We have made changes to the Manuscript, modified Tables and Figures, and tried to address the concerns and suggestions raised by the reviewer.
Specific comments
Comment 1: L 20 and throughout the manuscript: The term “parameters” is used frequently and in most cases more appropriate terms would be “variables” or “measures.” “Parameters” typically would be used to describe characteristics of variables such as means, standard errors, etc.
Answer: Thank you very much for this correction; the suggested change has been made throughout the Manuscript.
Comment 3: L 26, 37, 62, 64, 104, 112, 131, 142, 287, 294, and elsewhere as appropriate: Use commas after each term in a series of terms including after the penultimate term.
Answer: Thank you very much for your suggestion, which has been added throughout the entire text.
Comment 4: L 29 and elsewhere: The term “recipient” is not common from my perspective and “straw” should be substituted consistently throughout the manuscript.
Answer: Thank you very much for your suggestion, the suggested change has been made throughout the Manuscript.
Comment 5: L 45, 475, 485: Use “B(b)ecause” rather than “S(s)ince.”
Answer: Thank you very much for your suggestion; this has been changed in the text.
Comment 6: L 56: Delete “In addition to this” and start sentence with “Unfortunately, …” Answer: Thank you very much for your suggestion; this was changed in the text.
Comment 7: L 57-60: Too wordy, rewrite.
Answer: Thank you very much for your advice and suggestion. The referred text has been rewritten.
Comment 8: L 107: Include range of age of stallions if known.
Answer: Thank you very much for your suggestion; this has been included in the text.
Comment 9: L 274: Use plural “Statistical analyses”
Answer: Thank you very much for your advice and suggestion; this has been modified in writing.
Comment 10: L 286-297, L 349-353, Table 2 and Figure 3: A lot of effort went into defining and assigning individual spermatozoa into 4 subpopulations (SP1, SP2, SP3, and SP4) with SP3 and SP4 being defined but SP1 and SP2 were not specifically described. Regardless, there were no significant findings and Figure 3 is not needed; ranges of mean percentages with standard errors can be reported in the text. Table 2 may not be needed either but the data included may be of interest?
Answer: Thank you very much for your comment and suggestion. Since, as requested by one of the reviewer, the number of replications has been increased (from 7 to 13 animals), significant differences have been identified and are shown in Fig. 3. In addition, and given how the revised Figure 3 looks, we have advised that keeping Table 2 was required for the interpretation of this Figure.
Comment 11: L 301: Use “among” rather than “between.”
Answer: Thank you very much for your suggestion; this has been changed in the text.
Comment 12: Figure 1 includes no significant findings and ranges of percentages with standard errors could be simply reported in the text.
Answer: Thank you very much for your suggestion. The figure was removed from the text and added as supplementary material. In addition, and following the reviewer’s recommendation, examples of results (means with their standard errors) have been incorporated into the text.
Comment 13: Figure 2 includes only one significant result and that could be important or could be an outlier. Therefore, that Figure may not be needed either and the significant difference could be handled in the text and discussion.
Answer: Thank you very much for your suggestion. Because of the additional replicates conducted following the reviewer’s recommendation, this previously observed significant result is no longer present. Therefore, this Figure has been moved from the Manuscript body to Supplementary Material and examples of the results with their standard errors have been incorporated into the text.
Comment 14: Table 1 does not stand alone and not easily interpreted without referring to descriptions in the text. Each of the terms needs to be defined specifically in the table footnotes. Similar suggestion for Table 2 if it is kept.
Answer: Thank you very much for your suggestion. Following the advice of the reviewer, the definitions of each term used have been added to Table footnotes (table 1 and table 2).
Comment 15: Figures 4 and 5 need to include clear definition of terms used in the figure legends so they can be interpreted without referring to the text.
Answer: Thank you very much for your suggestion. We have analyzed this recommendation and that of the other reviewers regarding Tables and Figures, and legends have been revised in order to ensure that each term was clearly defined. However, since Figure legends were already very long, we have ensured that the revised ones complied with the Journal guidelines, which indicate that they must be explanatory but brief.
Comment 16: Figure 6 has no significant findings and those data can be reported as ranges with standard errors in the text.
Answer: Thank you very much for your suggestion, the figure was removed from the Manuscript body and added as supplementary material; however, examples of results (means) with their standard errors were incorporated into the text, as recommended by the reviewer.
Round 2
Reviewer 2 Report
This study (The effects of red light on mammalian sperm rely upon the color of the recipient and the medium used) is interesting and meaningful. The author has revised the article to improve the quality of the manuscript. The manuscript could be considered for publication in Animals.